# Advanced Therapy Medicinal Products for Age-Related Macular Degeneration; Scaffold Fabrication and Delivery Methods

**DOI:** 10.3390/ph16040620

**Published:** 2023-04-20

**Authors:** Hanieh Khalili, Hamid Heidari Kashkoli, David Edward Weyland, Sama Pirkalkhoran, Wiktoria Roksana Grabowska

**Affiliations:** 1School of Biomedical Science, University of West London, London W5 5RF, UK; 2School of Pharmacy, University College London, London WC1N 1AX, UK

**Keywords:** age-macular degeneration disease, artificial intelligence, cell-therapy, drug delivery, gene-therapy, 3D printing, retinal degeneration disease, scaffold

## Abstract

Retinal degenerative diseases such as age-related macular degeneration (AMD) represent a leading cause of blindness, resulting in permanent damage to retinal cells that are essential for maintaining normal vision. Around 12% of people over the age of 65 have some form of retinal degenerative disease. Whilst antibody-based drugs have revolutionised treatment of neovascular AMD, they are only effective at an early stage and cannot prevent eventual progression or allow recovery of previously lost vision. Hence, there is a clear unmet need to find innovative treatment strategies to develop a long-term cure. The replacement of damaged retinal cells is thought to be the best therapeutic strategy for the treatment of patients with retinal degeneration. Advanced therapy medicinal products (ATMPs) are a group of innovative and complex biological products including cell therapy medicinal products, gene therapy medicinal products, and tissue engineered products. Development of ATMPs for the treatment of retinal degeneration diseases has become a fast-growing field of research because it offers the potential to replace damaged retinal cells for long-term treatment of AMD. While gene therapy has shown encouraging results, its effectiveness for treatment of retinal disease may be hampered by the body’s response and problems associated with inflammation in the eye. In this mini-review, we focus on describing ATMP approaches including cell- and gene-based therapies for treatment of AMD along with their applications. We also aim to provide a brief overview of biological substitutes, also known as scaffolds, that can be used for delivery of cells to the target tissue and describe biomechanical properties required for optimal delivery. We describe different fabrication methods for preparing cell-scaffolds and explain how the use of artificial intelligence (AI) can aid with the process. We predict that combining AI with 3D bioprinting for 3D cell-scaffold fabrication could potentially revolutionise retinal tissue engineering and open up new opportunities for developing innovative platforms to deliver therapeutic agents to the target tissues.

## 1. Introduction

Age-related macular degeneration (AMD) is a progressive, multifactorial neurodegenerative disease of the macula and is a leading cause of irreversible blindness, affecting 1 in 8 people 60 years of age or older in the Western world [1]. AMD affects over 200 million individuals worldwide and is expected to affect close to 300 million people by 2040, which presents a significant public health concern and a substantial economic burden [1]. As global life expectancy increases, the socioeconomic burden of AMD is expected to become even greater in the future [2]. AMD is classified into three clinical stages of early, intermediate and late (or advanced) stage. The early and intermediate stages of the AMD disease are usually asymptomatic and characterised by the accumulation of insoluble extracellular aggregates called drusen (yellowish subretinal deposits made of proteins, lipids, and cellular debris), and the presence of abnormal pigments [3]. Late stage AMD is an advanced form of the disease that is characterised by geographic atrophy (GA) and neovascularisation [4].

AMD is further classified into two groups based on the absence or presence of neovascularisation: non-exudative or non-neovascular (‘dry’) AMD and exudative or neovascular (‘wet’) AMD [5]. Wet AMD is an advanced form of dry AMD which is characterised by the formation of fragile new blood vessels from the choroid into the retina. Several therapeutic options have been explored for the treatment of both wet and dry AMD, but only a few of them have entered clinical trials. These therapeutic approaches include antioxidant therapy, drug treatment targeting multiple pathways (angiogenesis, complement and inflammatory pathways), advanced therapy medicine (such as cell and gene therapy), as well as retinal implants [6].

The aim of this article is to provide an up-to-date review of recent progresses in development of advanced therapy medicinal products (ATMPs) for the treatment of AMD, including both scaffold and non-scaffold-based approaches. The properties of a scaffold such as porosity, permeability, biocompatibility, and fabrication methods have a major influence on the control of cell migration and growth [7]. Therefore, we aim to provide an overview of various methods and biomaterials that have been used in scaffold fabrication. Figure 1 shows an overview of scaffold fabrication and the delivery methods of cell-based and gene-based approaches for retinal degenerative disease. We will first describe the pathogenesis of AMD and available treatment options. We will then discuss different strategies for fabricating 3D scaffold-based cell therapies. We believe that 3D bioprinting offers superior benefits over other fabrication methods, and it is possible to apply artificial intelligence (AI) in scaffold optimization, which could eventually revolutionise retinal tissue engineering.

## 2. Pathogenesis of AMD and Treatment Options

Several factors are believed to play a role in the pathogenesis of dry AMD including genetics, oxidative stress, inflammatory [8], environmental [9] and ischaemic factors [10]. The formation of drusen is, however, considered as the hallmark for the earlier stages of dry AMD [11]. If drusen become enlarged and undergo confluence, it may transform into drusenoid which can cause detachment of the retinal pigment epithelium (RPE) (Figure 2) [12,13].

RPE is a monolayer of non-regenerative polygonal cells functioning as a barrier between the underlying Bruch’s membrane/choriocapillaris complex and the photoreceptors [15]. The RPE plays a crucial role in maintaining the integrity of photoreceptors (PR) and its dysfunction is implicated in a broad spectrum of degenerative retinal disorders such as AMD. The RPE cells keep retinal ganglion cells alive by facilitating the exchange of nutrients and water [16]. Photoreceptor cell death occurs as a result of separation of the photoreceptor cells from the underlying RPE cells and choroidal vessels [17] leading to irreversible blindness, given that the retina does not have endogenous stem cells to replace these damaged cells [18]. 

Drusen contains several proinflammatory factors such as complement components and tumour necrosis factors (TNF-α) [19]. Activation of complement pathway and the membrane attack complex (MAC) is suggested to play a critical role in the development of AMD and GA [20]. Elevated level of MAC in the outer layer of Bruch’s membrane has been identified as another hallmark of early stage AMD [21]. In addition, oxidative damage arising as a result of smoking, UV light exposure, oxidative stress, and mitochondrial damage to the retina, have all been suggested as key players in driving the progression of AMD [22]. Production of reactive oxygen species (ROS) and free radicals such as superoxide, hydrogen peroxide and hydroxyl radicals in the RPE lead to chronic oxidative stress and hence damage to the retina, however, the mechanisms have not been fully understood yet [23].

Accumulation of ROS results in overproduction of lipofuscin and β-amyloid [24]. A vicious circle is initiated where overexpression of inflammatory factors lead to increased production of reactive oxygen intermediates which reduce the bioavailability of antioxidants that are considered as a therapeutic option [25]. Anti-inflammatory agents such as immunosuppressants (e.g., corticosteroids or methotrexate), anti-TNF-α biologics and nonsteroidal anti-inflammatory drugs (NSAIDs) have also been considered as therapy options to inhibit progression of dry-AMD [26]. 

Furthermore, clinical studies have demonstrated that dietary antioxidants including zinc, carotenoids, flavonoids, and resveratrol, may serve as potential therapeutic interventions to prevent damage induced by ROS or even reverse vision loss [27]. As described in the age-related eye disease study (AREDS), a dietary intake with high levels of antioxidant vitamins and minerals, has the potential to decrease the risk of progression to advanced AMD in certain patients [28]. Results from several observational studies in AREDS suggest that increased dietary intake of omega-3 fatty acids and/or lutein and zeaxanthin is associated with an increase in macular pigment optical density [29]. 

Experimental evidence on different ω3 Polyunsaturated fatty acids (PUFAs) has documented the beneficial effects of docosahexaenoic acid (DHA) in modulating antioxidant gene expression in RPE cells under high glucose-like conditions [30]. In another study, DHA induced a potent antioxidant response by reducing lipases and β-oxidation enzymes and activating the Nrf2/Nqo1 signalling cascade, which is involved in the formation of reductive coenzyme NADH [31]. However, the AREDS2 randomized clinical trial suggested that adding DHA to the ARDES formula did not significantly reduce the progression of AMD [32,33].

Idebenone (IDB) has anti-apoptotic and cytoprotective effects on RPE cells under oxidative stress conditions, as confirmed by in vitro studies. These studies suggest that pre-treatment with IDB mediates the overexpression of Nrf-2 by increasing the level of Bcl-2 and decreasing the level of mitochondrial ROS levels, thus protecting against oxidative damage [34]. OT-551 is another novel molecule with antioxidant properties that protects against light-induced degeneration in RPE cells by downregulating the overexpression of the protein complex nuclear factor (NF)-kappa B [35].

The thickness of the Bruch’s membrane and the associated hypoxic conditions lead to overexpression of VEGF, which can give rise to angiogenesis and neovascularisation [34]. Currently, a variety of intravitreal anti-VEGF biologics, including monoclonal antibodies, antibody fragments, and bispecific antibodies have been used in the clinic for early stages of wet and/or neovascular AMD (nAMD) [36]. The approved anti-VEGF drugs include ranibizumab (Fab fragment, Lucentis^®^), aflibercept (Fc-fusion protein, Eylea^®^) and brolucizumab (scFV fragment, Beovu^®^) and faricimab (bispecific antibody consists of Fab fragment and modified Fc region, VabysmoTM). Bevacizumab, anti-VEGF full antibody, has also been used off-label to treat wet-AMD [37]. Although antibody-based medicines targeting VEGF, have revolutionised the treatment of neovascular (or ‘wet) AMD, anti-VEGF therapy is still far from perfect due to pharmacokinetic and compliance issues [38]. Currently, there is no effective treatment available to prevent or treat non-exudative (or ‘dry’) AMD [39].

To summarise, several therapeutic approaches are being investigated focusing on: (1) disease prevention, including antioxidant and visual cycle modulators, (2) halting disease progression, including drugs with anti-inflammatory, oxidative stress, mitochondrial enhancer, β-amyloid inhibitor and neuroprotective properties, and/or (3) vision restoration, including cell and gene therapy. While there are extensive reviews describing therapeutic options to address (1) and (2) in detail, which are listed in Table 1, we focus on vision restoration and the use of advanced therapy medicine in this review.

## 3. Advanced Therapy Medicines for Treatment of AMD

Although various pharmacological treatments are available for the treatment of AMD, the ideal treatment still does not exist. Advanced therapy medicinal products (ATMPs) are a group of complex and innovative biological products made for human use, consisting of cell therapy medicinal products (CTMPs), gene therapy medicinal products (GTMPs) and tissue engineered products (TEPs) [56]. The eye represents an ideal target for the application of ATMPs for several reasons. Firstly, it has relatively small dimensions, meaning that a small quantity of ATMP may be sufficient for effective treatment. Secondly, its compartmentalized anatomical structure can limit the distribution of therapeutic agents to non-target tissues. Thirdly, different administration methods to deliver ATMPs and examination of therapeutic outcomes is possible. Lastly, eye’s immunologically privileged status limits the movement of immune cells and molecules from the blood into the eye and aids in the development of ATMPS because it allows for the introduction of foreign substances (i.e., cells and gene) into the eye without triggering an immune response [56]. The field of ATMP development for ocular diseases is currently at the forefront of innovation as it offers the potential to identify novel therapeutic approaches for the treatment of eye conditions which were previously considered to be untreatable. Table 2 contains a summary of the cell- and gene-based approaches currently undergoing clinical trials for treatment of AMD. 

Cell therapy medicinal products are being developed to treat AMD by exploiting the immune-privileged environment of the subretinal space. Two mechanisms have been suggested, including replacing or regenerating the damaged RPE cells and introducing cells that can exert a supportive paracrine effect on photoreceptor function and survival [57,58]. Current research is focused on determining optimal transplantation targets (RPE, photoreceptors, choroidal endothelial cells), time and methods of administartion. Induced pluripotent stem cells (iPSCs), human embryonic stem cells (hESCs) and human umbilical cord (hUC)-derived cells are being used as cell sources [59]. Intravitreal injection of cell-based therapies has been studied, but the results have been inconclusive [60]. Sub-retinal injection of cell suspensions or cells cultured into a monolayer in vitro, has also been attempted as a method for delivering cells to the back of the eye. RPE cells injected in the form of a suspension have several drawbacks, such as tendency of RPE towards de-differentiation, rosette formation and efflux of cells into the vitreous cavity, problems that are not present in scaffold-based implants [59]. 

Transplantation of healthy RPE cells seeded in carefully crafted biomimetic scaffolds can better mimic morphology of native tissues and aid in the restoration of visual function [61]. The ideal scaffold will have to be non-immunogenic, mechanically robust but also sufficiently thin to allow exchange of nutrients and metabolites between the retina and choriocapillaris [61]. A scaffold-based approach can also enable cells to maintain a basal and apical polarisation via tight junctions before they are implanted [62]. Transplantations using various types of scaffolds have demonstrated enhanced cell survival and improved organisation of RPE cell populations [58]. A wide range of biomaterials have been used to engineer scaffolds for retinal tissue engineering. Although a scaffold-based approach may ensure better cellular performance in terms of physiology and cell survival (when compared to cell suspensions), some limitations remain. Delivery of RPE-scaffold materials into the subretinal space will necessitate the use of carefully designed instruments to minimise trauma [63]. In addition, formation of scars by microglia on various scaffolds has also been documented as a result of change in RPE cell’s phenotype and behaviour through the SMAD3 pathway [64]. Furthermore, several scaffolds have not been evaluated in vivo, and managing batch-to-batch variability and biomechanical properties can be challenging [61]. One significant challenge in retinal tissue engineering is establishing appropriate neural connections between the RPE implant and the native cellular environment. Recent developments in biomaterials science and stem cell research, as well as feedback from clinical trials, can assist in tackling the problems involved in creating clinical-grade CTMPs for AMD.

A number of clinical trials are currently underway to evaluate the effectiveness of cell-based therapy for the treatment of AMD. One such therapy is OpRegen^®^, in Phase1/2a for patients with progressive dry-AMD, which composed of RPE cells derived from human embryonic stem cells. The therapy is administered as a cell suspension in a single surgery to the subretinal space [65]. The primary goals of this study are to assess the proportion of patients in whom OpRegen^®^ can be delivered to the subretinal region, and to evaluate the safety of the procedure [65]. Another trial sponsored by the National Eye Institute, is investigating an autologous induced pluripotent stem cell (iPSC) therapy for the treatment of dry AMD. This approach involves generating iPSCs from somatic cells extracted from a patient with geographic atrophy, differentiating the iPSCs into RPE grown on a poly lactic-co-glycolic acid (PLGA) scaffold in vitro, and transplanting the RPE/scaffold construct into a small region in the subretinal space of the same patient, with the goal of rescuing the overlying neurosensory retina from further degeneration [66]. However, the results of these trials are yet to be published based on long-term assessments. Cell-based therapy has shown promise as an alternative treatment for preventing the progression of macular degeneration, but further trials are needed to determine their safety and efficacy. The use of hESCs for such therapies carries the risk of uncontrolled replication and teratoma formation [67]. Additionally, immunosuppressants are required to prevent implant rejection [68]. Human umbilical cord-derived cells (hUC) have limited differentiation capacity, while induced pluripotent stem cells (iPSCs) pose logistical challenges [38]. Surgical implantation procedures also carry an additional element of risk and there are safety concerns that need to be addressed. Assessing the effectiveness of transplants in cell therapy trials is a major challenge [67]. 

Gene therapy medicinal products (GTMPs) have also attracted the attention of researchers in the field of ocular therapeutics. Gene therapy involves introducing healthy genes to replace non-functioning or deficient proteins in patients’ cells using biological gene delivery vehicles or “vectors”. The therapeutic genes can be injected underneath the retina or directly into the vitreous body [69]. Gene-based therapy has the potential to offer sustained delivery of therapeutic agents in a single treatment [69]. For gene therapy to be effective, the vector must not cause an immune response or toxicity and must be able to sustain transgene expression [70]. Concerns about potential immune response to the vector used, are a significant issue in the development of GTMPs as it could negatively impact native tissues and lead to poor therapeutic outcomes. A variety of viral and non-viral vectors have been used in gene therapy; recombinant viral vectors remain the preferred delivery vehicledue to their stability and therapeutic efficacy [71]. However, the use of lentivirus-mediated gene therapy is limited due to the risk of insertional mutagenesis and poor transduction of retinal cells [72]. Recombinant adeno-associated viral vectors (AAVs) are widely recognised as versatile vectors for gene-based therapy, especially for retinal gene supplementation. They have a prolonged expression profile and high efficiency in transduction of multiple cell types [57], with low immunogenicity and vector-related toxicity [57]. Unlike lentiviral vectors, AAVs are predominantly non-integrating [73] and their small, single-stranded DNA genome of approximately 4.7 kb and capsid organisation greatly facilitate genetic modification, thereby enabling customisation of their properties [70]. AAVs are also available in multiple serotypes and their genome can persist as an episomal concatemer in transduced cells, resulting in long-lasting expression of the transgene in non-dividing retinal cells [57]. Early-phase gene therapy clinical trials for AMD using AAVs have shown promising results, and more clinical trials are expected in the near future.

A biotechnology company called SparingVision is currently developing SPVN06, an AAV-based gene therapy product that could potentially counteract the degeneration of cone photoreceptors in dry age-related AMD by restoring RdCVF, a neurotrophic factor secreted by photoreceptor cells in the retina, and by promoting RdCVFL, a potent antioxidant which protects retinal cone cells against damage caused by oxidative stress [74]. SparingVision has recently obtained FDA clearance for its IND application for SPVN06 and has also submitted a clinical authorisation application to the French regulator (ANSM). Several other gene therapies are also currently undergoing clinical trials. Phase 1 of a clinical trial for a gene therapy called HMR59, sponsored by Janssen Research & Development LLC, was recently completed for the treatment of both wet and dry AMD. This gene therapy can permanently alter retinal cells to upregulate expression of a soluble form of CD59. The soluble recombinant version of CD59 is capable of protecting retinal cells from the destructive effects of membrane attacking complex (MAC) [75]. Another example of an ongoing gene therapy trial for AMD is GT005, which is currently being sponsored by Gyroscope Therapeutics. Similar to HMR59, this therapy also targets MACs, however it relies on upregulation of complement factor I (CFI) protein to stabilise an overactive complement system [76]. 

Whilst gene therapy represents an ideal option for treating a debilitating condition like AMD, a number of challenges still remain in identifying therapeutic targets and designing efficient gene delivery vectors. The route of vector administration is also a major determinant of the efficacy of gene therapies [70]. Three principal modes of delivery are routinely used, namely intravitreal, subretinal and suprachoroidal. Although the overwhelming majority of research groups utilise AAV vectors, a number of concerns have been raised over their ability to successfully transduce cells after intravitreal injection [77]. Although intravitreal injection is routinely used in clinical practice to deliver anti-angiogenic agents to treat neovascularisation associated with wet AMD, the majority of currently available AAVs have poor rates of penetration into the outer retina when administered intravitreally [72]. Subretinal delivery may be more effective for diseases affecting the RPE, however, it carries additional risks such as cataract formation, retinal damage and potentially vision loss [75]. Subretinal injection can separate the photoreceptor layer from the supporting RPE layer, leading to impaired function and reduced rate of survival even in healthy retinas [76]. Furthermore, overall effectiveness may be constrained by the fact that subretinally injected AAV vectors only transduce a low proportion of outer retinal layer cells that are closely in contact with the subretinal bleb [76]. It must also be taken into account that the current capacity of AAV vectors is limited to 4.7 kb, unless a dual/triple vector is employed [77]. Suprachoroidal delivery has also been investigated; in this method the therapeutic agent is injected into the suprachoroidal space between the choroid and the sclera [72]. This approach holds great promise for less invasive delivery to retina, however there are concerns about undesirable immune responses and penetration through multiple layers of tissue [72]. 

Several early-phase clinical trials have shown promising results, however, translating gene and cell therapies from the laboratory to the clinic remains a significant challenge. While gene therapies could potentially have a substantial impact on the treatment of AMD, their long-term effectiveness has been hindered by the body’s response and issues related to ocular inflammation which could lead to vision loss. Currently, AAV-based gene therapies are limited to targeting long-lived cells [73]. Moreover, evaluating AAV vector performance using in vitro models is not ideal, and results obtained from small animal models may not be reproducible in human trials [78]. Bioprocessing and economical production of gene and cell therapies with adequate critical quality attributes also pose major challenges to getting such products to patients and realising the full potential of advanced therapy medicinal products. 

**Table 2 pharmaceuticals-16-00620-t002:** List of ATMPs currently undergoing R&D and clinical trials for treatment of AMD. Cell-based therapy includes both scaffold-based and scaffold-free approaches. While subretinal route is the main route of administration for both cell and gene therapy, other delivery methods have also been investigated.

Methods		Product Name	Study Phase	Therapeutic Agent (s)	Method of Delivery
**Cell-Based Therapy**	Scaffold-free	OpRegen [65]	Phase IIa	Human embryonic stem cell (hESC)-derived RPE cells	Subretinal administration as a cell suspension either in ophthalmic Balanced Salt Solution Plus (BSS Plus) or in CryoStor^®^ 5 (Thaw-and-Inject, TAI)
		RPESC-RPE-4W [79]	Phase I/IIa	Allogeneic RPE stem cell (RPESC)-derived RPE cells	Subretinal administration; RPESC-RPE cell obtained after 4 weeks of differentiation (RPESC-RPE-4W). The RPESC-RPE-4W progenitor stage cell has shown increased engraftment and vision rescue compared to more mature RPE cell products
		AlloRx [80]	Phase I	Cultured allogeneic adult umbilical cord derived mesenchymal stem cells	Intravenous and sub-tenon administration; It has the potential to reduce inflammation through activation of anti-inflammatory biochemical and cellular pathways
	Scaffold-based	iPSC-derived RPE/PLGA transplantation [66]	Phase I/IIa	iPSC-derived RPE	Subretinal administration; iPSCs are differentiated into RPE, which is grown as a monolayer on a thin poly lactic-co-glycolic acid (PLGA) scaffold
		CPCB-RPE1 [81]	Phase I/II	Human embryonic stem cell (hESC)-derived RPE cells	Subretinal administration; Implant is designed to replace the RPE and Bruch’s membrane in the eye that degenerate in AMD
		PF-05206388 [82]	Phase I	Human embryonic stem cell derived retinal pigment epithelium	Subretinal administration; Monolayer of RPE cells immobilized on a polyester membrane It is a living tissue equivalent, which is designed to remain in situ life-long

**Gene-Based Therapy**		ADVM-022 [83]	Phase I	AAV.7m8 gene vector carrying a coding sequence for aflibercept	Intravitreal administration: One-time IVT administration of ADVM-022 provides durable expression of therapeutic levels of intraocular anti-VEGF protein (aflibercept)
		FT-003 [84]	Phase I	AAV vector	Intravitreal administration FT-003 has the potential to treat AMD by providing durable expression of therapeutic levels of intraocular protein
		4D-150 IVT [85]	Phase I/II	AAV-based gene therapy carrying miRNA targeting VEGF-C and codon-optimized sequence encoding aflibercept	Intravitreal administration: Dual-transgene gene therapy designed to inhibit four distinct angiogenic factors to prevent angiogenesis and reduce vascular permeability.
		BD311 [86]	Phase I	Integration-deficient lentiviral vector (IDLV) expressing VEGFA antibody	Suprachoroidal administration: Gene is delivered to the RPE cells to express the VEGFA antibody to neutralizes the VEGFA activity in the posterior segment
		RGX-314 [87]	Phase II	AAV8 vector that contains a gene to encode for a monoclonal antibody fragment to neutralizes VEGF	Subretinal administration: RGX-314 is being developed as a potential one-time treatment for wet AMD

## 4. Fabrication Strategies for Scaffold-Based Retinal Implants

Although cell and gene therapies have demonstrated great promise, their translation from bench to bedside remains a significant challenge. Tissue engineering (TE) offers an alternative approach for treating AMD by developing tissue engineered products (TEPs) that can replace dysfunctional RPE and restore its function, thus halting the progression of the disease [88]. TE for RPE regeneration requires scaffolds with suitable properties that allow cellular interactions, proliferation, and differentiation to take place, whilst preventing undesirable host immune responses and inflammation. Designing a 3D scaffold with adequate structural parameters and bioactivity is integral to achieving this goal [89]. Intuitively, the scaffold should be able to mimic the native extracellular matrix (ECM) activity of target tissues, provide structural support and a temporary matrix [90]. Table 3 provides a summary of the most important properties of an ideal scaffold: biocompatibility, biodegradability, tuneability and self-healing properties [91]. Scaffolds may be seeded with different types of cells (e.g., stem cells, progenitor cells, differentiated cells, etc.) or can also be implanted directly to guide functional tissue regeneration in dysfunctional tissues. From an engineering perspective, it is essential that scaffold properties and their regulatory functions are integrated within scaffold design in a reproducible manner, so that tissue regeneration can be induced at the site of implantation [92]. Due to the highly complex nature of biological systems, replicating natural processes and bioengineering tissues with the aid of a scaffold is a challenging task.

A thorough understanding of fundamental scaffold characteristics and functions is crucial for developing an effective scaffold fabrication strategy. The type of biomaterial used for scaffold fabrication is another important parameter in scaffold design. The selection of biomaterials plays a key role in determining the biomechanical properties of the scaffold and its ability to interact with native tissues in a biologically appropriate manner, rather than acting as an ‘inert body’ [101]. Natural, syntheticand hybrid biomaterials have been used to fabricate scaffolds for retinal tissue engineering. Natural biomaterials such as collagen, gelatin, alginate and hyaluronic acid [61] are commonly used. Synthetic materials including PCL, PDLLA PLGA, PTMC Parylene-C, have also been explored [62]. Decellularised tissues and thermo-responsive polymers are being investigated as well [102]. However, selection an appropriate fabrication technology for engineering 3D scaffolds remains a significant challenge [103].

Various techniques have been developed for manufacturing 3D scaffolds, including electrospinning, freeze-drying and 3Dbioprinting [104]. The electrospinning technique has the advantage of providing an environment with fibre diameters down to the nanometre scale to facilitate cell attachment. Electrospinning requires a syringe pump, electrostatic force, and metallic collector to generate fibres [105]. As shown in Figure 3, a very high voltage is applied simultaneously to a metallic needle filled with polymeric solution that is placed at the tip of the syringe via surface tension. The electric field and the potential difference between terminals result in the repulsion of electric charges, followed by drawing out of polymers and deposition onto a collector [105]. A variety of synthetic and natural 3D nanofibrous polymers, such as a biomimetic nanofibrous scaffold comprising type I collagen with 1,1,1,3,3,3-hexafluoro-2-propanol, have been fabricated using the electrospinning method [106]. Despite its advantages, electrospinning still has some major limitations such as inadequate mechanical properties, use of toxic solvents, poor control over cell density, insufficient cell infiltration and non-homogenous cell distribution [107].

The freeze-drying (lyophilization) method is another commonly used technique in regenerative medicine for creating water-soluble polymer scaffolds through the process of sublimation [110]. The process of freeze-drying consists of three main stages: (1) pre-freezing, which involves forming an interpenetrating network of ice crystals by freezing a suspension of the water-soluble polymer at extremely low temperatures in the range of −70 °C to −80 °C, (2) primary drying, during which the bulk of ice crystals is removed under low pressure by sublimation (3) secondary drying, which involves the extraction of unfreezable, bound and associated water via a desorption process, leading to the formation of a highly porous scaffold (Figure 3) [111]. Freeze-drying is a convenient and simple technique to fabricate porous cell scaffolds, however, construction of a suitable scaffold shape with controllable microarchitecture from natural biomaterials remains a significant challenge with this method. In addition, the high rate of rehydration during freeze-drying process could lead to changes in cell diameter due to swelling of the cell wall, resulting in dramatic structural changes [112].

More recently, 3D bioprinting has shown great potential in tissue engineering for scaffold fabrication, by integrating living cells (bioink) and biomaterials directly or indirectly together using layer-by-layer approach [113]. To date, 3D bioprinted constructs have mainly been classified into two forms: acellular and cellular constructs. In acellular bioprinting, scaffold fabrication occurs without cellular material, which is only incorporated after the printing process is complete. In contrast, cellular bioprinting involves incorporating cells and other biological agents within the biomaterial during the fabrication process [114]. The bioink is a critical aspect of 3D bioprinting, as it must be highly biocompatible and bioprintable [108]. Three distinct approaches have been used for fabricating artificial Bruch’s membrane in TE using various 3D bioprinting methods: extrusion-based (EBB), droplet-based and laser-assisted bioprinting [111]. Compared to other techniques, EBB benefits from greater versatility, less process-induced cell damage, and the ability to precisely deposit very high cell densities through computer-aided design (CAD), making it ideal for printing more geometrically complex scaffolds with ease of customisation [115].

## 5. Conclusions and Future Perspective

The human retina is a highly complex tissue that serves as a key component of both the eye and central nervous system. Any damage to retinal cells including photoreceptor cells, can lead to permanent vision loss because the retina lacks an endogenous stem cell population to replace degenerated and/or damaged cells. AMD is a retinal degenerative disease that affects around 20 million people in the US alone, and it is projected that approximately 300 million worldwide will have the disease by 2040. Although treatment options exist for the neovascular form of AMD to prevent and halt progression of the disease, opportunities for restoring vision are limited once photoreceptor cells become dysfunctional. Advanced therapies represent an emerging field of research which seek to develop cell- and gene-based therapies to treat and potentially cure degenerative diseases such as AMD. Different products for both cell- and gene-based approaches are currently in clinical trials for treatment of both AMD and GA, however, no product has so far reached the market and received FDA approval. This must be due to challenges in selecting the appropriate gene and/or cell types, designing suitable delivery vehicles and ensuring ocular tolerability. Natural polymers also have their own limitations in terms of processability and mechanical properties. Changes in the constituents of natural polymers can lead to accumulation of debris in Bruch’s membrane and eventual cell death.

Recently, scaffold-based approaches for treating retinal degenerative diseases have provided a promising avenue to halt or prevent disease progression in AMD. Nevertheless, different biomaterials and methods of scaffold fabrication can have a significant impact on product success. The freeze-drying method is a primary step that is simple and convenient for making porous structures but has limitations such as (i) poor reproducibility, (ii) poor mechanical properties and (iii) less controllable microstructures and properties. Electrospinning technology has the ability to control fibre diameter and provide a high surface area for cell attachment. However, this method requires the use of toxic organic solvents during processing which can have an adverse effect on cell viability. 3D-bioprinting represents a more attractive scaffold fabrication method with potential advantages of superior physical characteristics, cell adhesion, better control of scaffold microstructures and overall improved mechanical properties, compared to freeze drying.

Multi-objective optimisations are, however, required to identify suitable printing conditions for 3D bioprinting (e.g.: print speed, pressure, layering and spacing) and material composition (percentage of cells to bioink) to achieve optimal mechanical and porous constructions properties. This requires extensive experimentation time that is resource-demanding. Artificial intelligence (AI) and machine learning (ML) methods have already been applied to solve problems in scientific research and represent potentially transformative resources to support researchers in the field of regenerative medicine. In the context of 3D bioprinting, an ML-based supervised learning framework could take the material composition and the printing parameters as input to (i) to enable optimisation of scaffold properties, (ii) assess the quality of the prints and (iii) optimise printability of the material. Given the appropriate training datasets, a machine learning algorithm could generate an accurate prediction model for efficient scaffold fabrication and provide a quantitative evaluation of scaffold printability for specific anatomical sites, such as the human retina. We predict that combining AI with 3D bioprinting for 3D cell-scaffold fabrication will revolutionise retinal tissue engineering and open up a world of new opportunities for developing novel drug delivery platforms for the treatment of retinal diseases.

## Figures and Tables

**Figure 1 pharmaceuticals-16-00620-f001:**
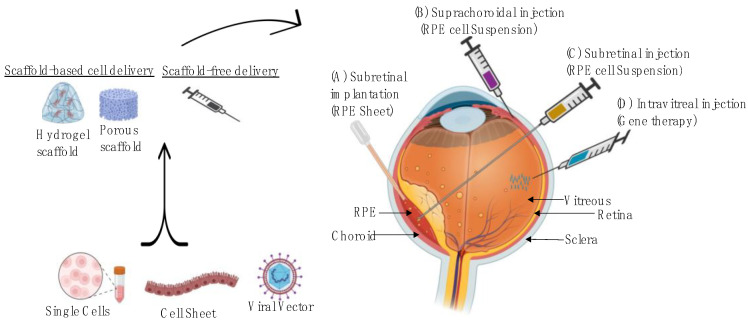
Overview of scaffold fabrication and the delivery methods of cells-based and gene-based approaches for the retinal degeneration disease. (**A**) Subretinal implantations, performed in the operating room, allows for direct delivery of RPE sheet as a cell suspension into the subretinal space, (**B**) Suprachoroidal injection is a nonsurgical method to deliver therapeutic suspension into a virtual space between the sclera and choroid, (**C**) Subretinal injection delivers RPE cell suspension under the sensory retina using a micro-needle and (**D**) Intravitreal injection method is applied in an office setting to deliver vectors to the vitreous cavity.

**Figure 2 pharmaceuticals-16-00620-f002:**
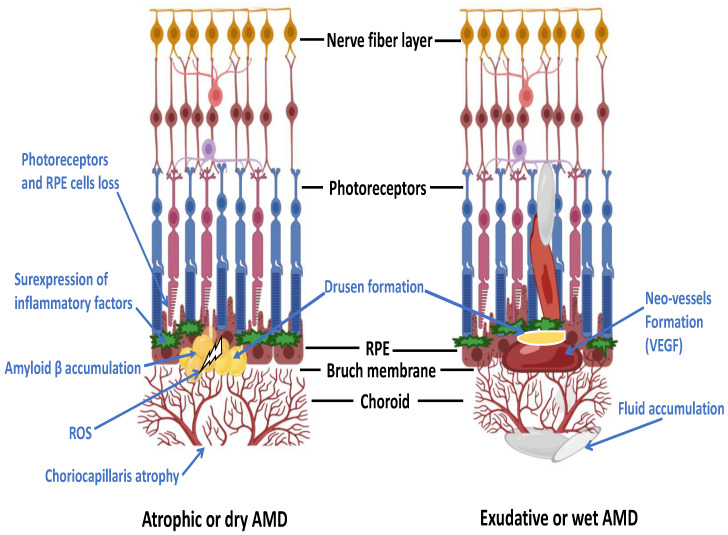
Diagram showing the structure of the retina, the RPE and choroid in the dry and wet forms of AMD. Late AMD is characterised by drusen formation, neovascularisation, disruption of Bruch’s membrane as well as RPE cell damage and death [14].

**Figure 3 pharmaceuticals-16-00620-f003:**
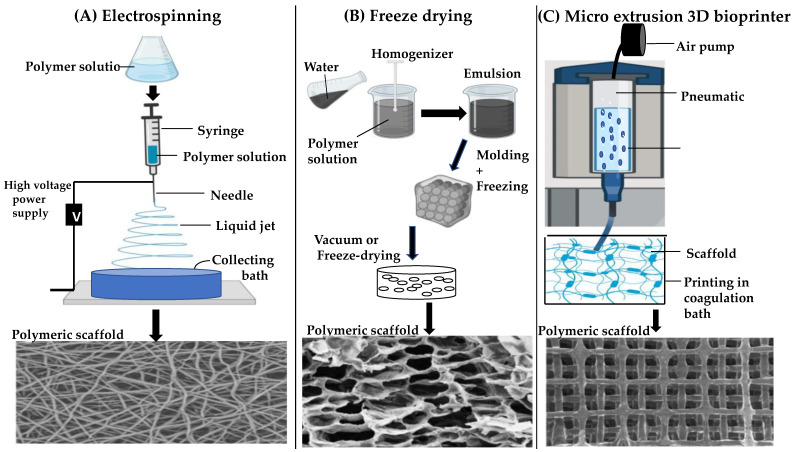
Schematic illustration of different techniques for manufacturing 3D tissue scaffold and optical, SEM images of the printing constructs [108,109,110].

**Table 1 pharmaceuticals-16-00620-t001:** List of potential treatment options in clinical trials for dry-AMD, focusing on disease prevention and halting disease progression.

Disease Prevention	Product Category	Product Name	Study Phase	Mechanism of Action	Method of Delivery
	Antioxidant	AREDS [28]	Phase III	vitamin supplement	Oral
		OT-551 [40,41]	Phase II	vitamin supplement	Eye drop
	Visual cycle modulators (Prevent drusen formation)	ACU-4429 [42]	Phase II/III	Inhibits the formation of 11-cis-retinal to slow the rate of retinoid metabolism and A2E generation	Oral
		Fenretinide [43]	Phase II	Synthetic retinoid (vitamin A); reduce accumulation of lipofuscin through binding to its carrier protein	Oral
		C20-D3-vitamin A (ALK-001) [44]	Phase III	A modified form of Vitamin A to decrease toxic by-product formation through reducing A2E biosynthesis	Oral

Halting Disease Progression	Anti-inflammatory drugs (anti-complement pathways)	Eculizumab [45]	Phase III	A monoclonal antibody to inhibit the complement protein C5, preventing MAC formation	IV
		Lampalizumab [46]	Phase III	A monoclonal antibody to inhibit complement factor D	Intravitreal
		Avacincaptad pegol (Zimura) [47]	Phase II/III	Anti-complement factor 5, preventing MAC formation	Intravitreal
		Pegcetacoplan (APL-2) [48]	Phase II	Complement C3 inhibitor and prevents downstream activation of C3b	Intravitreal
		LFG316 [49]	Phase II	A monoclonal antibody to inhibit the complement protein C5	Intravitreal
	Oxidative stress	Risuteganib [50]	Phase II	An integrin inhibitor of αVβ3/αVβ5 and α5β1 to target multiple oxidative stress factors	Intravitreal
	Mitochondrial enhancer	Elamipretide [51]	Phase III	A small mitochondrially targeted tetrapeptide to reduce the production of toxic ROS and stabilize cardiolipin levels	Subcutaneous
	β-amyloid inhibitors	GSK933776 [52]	Phase II	An anti-amyloid β monoclonal antibody	IV
		RN6G [53]	Phase II	A humanized antibody to inhibit accumulation of amyloid β-40 and β-42	IV
	Neuroprotection	Ciliary nerve trophic factor [54]	Phase II	Protects rod photoreceptors and retinal cones by improving morphology of photoreceptor mitochondria and reduingoxygen consumption	Intravitreal
		Brimonidine tartrate [55]	Phase II	An alpha2-adrenergic receptor agonist	Intravitreal

**Table 3 pharmaceuticals-16-00620-t003:** Desirable characteristics of a tissue-engineered scaffold.

**Scaffold Property**	**Biological Significance**	**Refs**
Biocompatibility	(1) Provide a normally functioning matrix to enable cell adhesion, proliferation and ECM generation (2) Maintain cellular viability within scaffold and at the site of implantation (3) Following implantation, TE construct must not give rise to adverse immunological response (e.g., inflammation, fibrotic scarring, immune rejection, impaired healing)	[93,94]
Biodegradability	(1) Regeneration of fully functional tissue should coincide in time with the complete degradation and resorption of the scaffold (2) The rate of degradation should kinetically match with the changing microenvironment during the regeneration process (3) Biodegradation can have a critical role in the provision of pathways for angiogenesis and metabolite diffusion, as well as release of therapeutic agents loaded into the scaffold	[95,96,97]
Mechanical properties	(1) Scaffold must fulfil key mechanical functions of tissue that is being replaced (2) Mechanical properties should ideally be consistent with implantation site to enable mechanotransduction and minimise native tissue damage (3) Scaffold should have sufficient mechanical integrity to allow surgical handling during implantation (unless delivered by injection)	[96,97]
Scaffold architecture	(1) Scaffold should mimic the architecture of the native tissue (e.g., Bruch’s membrane) as closely as possible to allow remodelling to take place (2) Architecture should display sufficient interconnectivity and porosity to allow the transfer of nutrients/waste products and facilitate adequate cellular penetration and migration	[98,99,100]
Manufacturing technology	(1) The ideal scaffold needs to be manufacturable to GMP standards (2) Amenable to scalable and cost-effective bioprocessing	[100]

## Data Availability

Data sharing not applicable.

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
