# Peer review of "Advanced Therapy Medicinal Products for Age-Related Macular Degeneration; Scaffold Fabrication and Delivery Methods"

_pharmaceuticals, 2023, doi:10.3390/ph16040620_

Round 1
Reviewer 1 Report
The review article titled, Recent advanced therapy medicine for age macular degeneration is about the advanced ATMPs approaches including cell and gene based therapies for the treatment of dry AMD along with their applications.
The title of the article is appealing but the data presented is not sufficient and is not well organized. Author have described very short information about the ATMPS and Gene based therapies, moreover the delivery vehicles are also not described with sufficient details to grasp the readers attentions and provide a comprehensive information about the topic.
All the topics covered in this review have very basic information and according to the scope of the journal this article does not seems appropriate for publication.
Author Response
We would like to thank the Reviewer for the comments. We have extensively edited the English and style of the revised manuscript. Additionally We have thoroughly edited and updated each section in accordance with the Reviewer’s recommendations. Specifically, we have added additional information on; (i) the pathogenesis of wet AMD, (ii) the delivery method for ATMPs, accompanied by a new figure (figure 1 in the revised manuscript) in the introduction, and (iii) tissue engineering strategies for AMD to provide a more comprehensive overview on the topic. We have completely revised and re-organised Table 3. In our revision, we have also added 38 more references to support the additional information. Furthermore, we have slightly changed the title to better reflect the content presented in the review article.
Reviewer 2 Report
The authors present an interesting review on the therapeutic approaches to AMD including most recent therapeutic strategies.
My only concern is the following: The section related to antioxidants is somehow neglected: authors should be clearer on the difference between preventive and therapeutic approaches with antioxidants, Including DHA, Idebenone, punicalagin flavonoids and other active principles. They should refer to The following works:
DHA
Antioxidants 2022, 11(6), 1072; https://doi.org/10.3390/antiox11061072
Antioxidants 2023, 12(2), 339; https://doi.org/10.3390/antiox12020339
Idebenone
Biomedicines 2022, 10(2), 503; https://doi.org/10.3390/biomedicines10020503
Flavonoids
Investigative Ophthalmology & Visual Science July 2006, Vol.47, 3164-3177.
Author Response
We would like to thank the Reviewer for careful reading of the manuscript and putting our work into the context. We have extensively edited the English in the revised manuscript. We haveadded further details on the antioxidants in the revised manuscript as requested. Specifically, we have focused on flavonoids, ARDES, DHA and idobenon (IDB) as potential protective agents for the progression of AMD. We have also added the recommended references to support these additions in the revised manuscript.
Reviewer 3 Report
1. In the abstract section, quantitative data must be included.
2. Please give a "take-home" message as the conclusion of your abstract.
3. Rearrange keywords alphabetically.
4. Keywords needs to provided.
5. It is encouraged not to use abbreviations in the keywords section.
6. The novelty in the current article by the authors is too weak. The past has seen extensive published work of written material. It is required to provide more details for more explanation about the present novel in the introductory section.
7. It is essential to summarize previous works' merits, novelties, and limitations in the introductory part to emphasize the gaps in the research that the latest research seeks to address.
8. Line 51, before state the present review objective related scaffold. The role and basic concept of scaffold needs to explained. For this purpose, please refer the previous relevant study as follows: The Effect of Tortuosity on Permeability of Porous Scaffold. Biomedicines 2023, 11, 427. https://doi.org/10.3390/biomedicines11020427
9. The current submission needs to make proper editing in the objective of the present work in the last paragraph of the introduction section.
10. Additional figures in the introduction would improve the quality of the present article. Please provide it.
Author Response
We are grateful for the Reviewer comments. While, we have addressed the specific comments raised by the Reviewer in the attached letter, we have also extensively edited the English and style of the revised manuscript during the revision process.

Round 2
Reviewer 1 Report
Authors have tried best efforts to improve the content and quality of the review article.
but still there are some querries, which must be addressed before the article could be accepted.
There are many English mistakes and repetition of some words.
Font style of figures captions and text is not same.
Overall article is in better form.
Author Response
Comments: Authors have tried best efforts to improve the content and quality of the review article. but still there are some queries, which must be addressed before the article could be accepted.
- There are many English mistakes and repetition of some words.
Response: We thank the Reviewer for this comments. We have extensively edited the document and proofread the text during second round of revision to ensure that all of the typos and mistakes have been corrected.
- Font style of figures captions and text is not same.
Response: We have edited the font style in the text, figures and table during second round of revision. We also edited the table style to ensure all the tables are in the same format.
Overall article is in better form. THANKS
Reviewer 2 Report
My concerns are addressed. best regards
Author Response
We thank the Reviewer for approving our revision
Reviewer 3 Report
Following comments is given as follows:
1. Instead of only using the dominating text as a present form, the authors should also include extra illustrations in the form of figures that clarify the workflow of the current study to make the reader's understanding simpler.
2. An evaluation of the findings with similar past investigations is required.
3. One of the important aspects in scaffold in porosity, please explain it in brief and incorporated relevant reference as follows: Level of Activity Changes Increases the Fatigue Life of the Porous Magnesium Scaffold, as Observed in Dynamic Immersion Tests, over Time. Sustainability 2023, 15, 823. https://doi.org/10.3390/su15010823
4. The authors need to improve the discussion in the present article to become more comprehensive. The present form was insufficient.
5. The limitation of the current article must be included at the end of the discussion section.
6. Provide a paragraph-length conclusion rather than the present form's point-by-point description.
7. Please explain the further research in the conclusion section.
8. In the entire manuscript, the authors occasionally constructed paragraphs with just one or two phrases, which made the explanation difficult to understand. To make their explanation a full paragraph, the authors should expand it. It is advised to use at least three sentences in a paragraph, with the primary sentence coming first and the supporting sentences coming after.
9. Five years back literature should be enriched into the reference. MDPI reference is strongly recommended.
10. Please reduce the literature used as a reference that is authored by the present author in order to reduce the number of self-citation.
11. Due to grammatical problems and linguistic style, the authors should proofread the work.
12. Provide graphical abstract for submission after revision.
